# Viscoelastic Memory Effects in Cyclic Thermomechanical Loading of Epoxy Polymer and Glass-Reinforced Composite: An Experimental Study and Modeling Under Variable Initial Stress and Cycle Durations

**DOI:** 10.3390/polym17030344

**Published:** 2025-01-27

**Authors:** Maxim Mishnev, Alexander Korolev, Alexander Zadorin, Daria Alabugina, Denis Malikov, Fedor Zyrianov

**Affiliations:** Department of Building Construction and Structures, South Ural State University, Chelyabinsk 454080, Russia; zadorinaa@susu.ru (A.Z.); alabugina_darya@mail.ru (D.A.); malikovda@susu.ru (D.M.); zyrianovfa@susu.ru (F.Z.)

**Keywords:** epoxy polymers, glass reinforced plastics, viscoelasticity, structural model, thermal stresses, residual stresses

## Abstract

This article presents a study of the viscoelastic behavior of an epoxy polymer and a glass-reinforced composite based on it under cyclic thermomechanical loading. The goal is to model and explain the experimentally observed stress state formation, including the accumulation of residual stresses under various initial mechanical stress levels and heating/cooling cycle durations. An improved material model, implemented as a Python script, is used, allowing for the consideration of memory effects on thermomechanical loading depending on the level and nature (mechanical or thermal) of the initial stresses. A Python script was developed to determine the viscoelastic parameters (elastic modulus E_1_, elastic parameter E_2_, and viscosity) for the three-element Kelvin–Voigt model. These parameters were determined at different temperatures for both the polymer and the glass-reinforced composite used in the modeling. The accumulation of stresses under different ratios of mechanical and thermal stresses was also investigated. Experiments showed that high levels of residual stress could form in the pure epoxy polymer. The initial stress state significantly influences residual stress accumulation in the pure epoxy polymer. Low initial tensile stresses (0–1.5 MPa) resulted in substantial residual stress accumulation, exceeding the initial stresses by up to 2.7 times and reaching values of up to 2.1 MPa. Conversely, high initial stresses (around 4 MPa) suppressed residual stress accumulation due to the dominance of relaxation processes. This highlights the critical role of the initial loading conditions in predicting long-term material behavior. In the glass-reinforced plastic, the effect of residual stress accumulation was significantly weaker, possibly due to the reinforcement and high residual stiffness, even at elevated temperatures (the studies were conducted from 30 to 180 °C for the composite and from 30 to 90 °C for the polymer). The modeling results show satisfactory qualitative and quantitative agreement with the experimental data, offering a plausible explanation for the observed effects. The proposed approach and tools can be used to predict the stress–strain state of polymer composite structures operating under cyclic thermomechanical loads.

## 1. Introduction

Polymer composites are increasingly used in diverse industries, including civil engineering, for applications ranging from protective coatings and insulation to load-bearing structural elements. Their high corrosion resistance, favorable strength-to-weight ratio, and other advantageous properties make them well-suited for demanding service environments, such as industrial gas exhaust systems (chimneys, ducts) [1,2,3,4], gas purification and desulfurization installations, storage tanks, pressure vessels, and pipelines [5,6,7,8,9]. These structures often operate under combined exposure to aggressive environments, cyclic temperature variations, and mechanical loads. Understanding the behavior of polymers and polymer composites under these complex conditions is crucial for accurately predicting their stress–strain state (SSS), load-bearing capacity, and long-term reliability.

This study focuses on generating novel experimental data and developing methods for designing more reliable and cost-effective polymer composite structures for such industrial applications. By investigating operational residual stress accumulation, this work aims to expand the application range of these structures. Beyond reducing repair and operational costs, the inherent corrosion resistance of polymer composites enables the integration of enhanced gas cleaning and heat recovery technologies, contributing to improved environmental performance.

The viscoelastic thermomechanical behavior of polymers and composites, including residual stress formation, has been extensively researched using various constitutive approaches [10,11,12,13,14,15,16,17,18]. However, predicting the long-term performance of polymer composite structures under cyclic thermomechanical loading remains challenging, as evidenced by recent publications. This research spans diverse materials and applications, from interfacial interactions in heterostructures [19] and strain effects [20] to the influence of cyclic temperatures, aggressive environments (including seawater), and mechanical loading on polymer composites, including residual stress development, damage, and failure [21,22]. The formation of residual stresses can be influenced not only by temperature variations but also by other factors, such as swelling due to water absorption, especially in aggressive environments. However, this work focuses specifically on the influence of cyclic thermomechanical loading on the accumulation of residual stresses in polymers and polymer composites used in long-term civil engineering applications, which requires further investigation and remains a key focus of current research. While the proposed approaches are expected to be applicable to a wide range of polymers and composites, their validity is demonstrated in this study using a specific type of thermosetting resin and a composite based on it.

The mechanical behavior of polymers and composites is a complex nonlinear phenomenon influenced by multiple factors, such as production technology [23], geometric features [24,25], material composition and microstructure (including effects occurring in the fiber-matrix interface), temperature-time history, loading conditions, and internal processes such as creep, relaxation, and stress redistribution. A key consequence of constrained deformation during heating and cooling cycles is the development of residual stresses, which remain after load removal and can accumulate under cyclic conditions. These stresses arise from constrained thermal deformations and evolve through interconnected creep and relaxation processes, the rates of which are temperature- and load-dependent. Residual stress accumulation can significantly impact structural durability and operational characteristics. For instance, in shell structures like industrial gas exhaust tracts, tanks, and pipelines, accumulated residual tensile stresses can lead to cracking and loss of integrity, particularly in unreinforced or lightly reinforced polymer components [26,27], such as protective chemically resistant layers, potentially causing structural failure (Appendix B, Figure A1).

Accurate prediction of the long-term SSS of load-bearing polymer composite structures designed for decades of service is therefore essential. This requires accurate prediction of long-term (rheological) deformation and a thorough understanding of residual stress influence on structural load-bearing capacity [28,29,30]. Even seemingly insignificant short-term effects, such as gradual operational residual stress accumulation, can have serious long-term consequences, necessitating careful evaluation.

The viscoelastic thermomechanical behavior of polymers and composites and residual stress formation have been widely studied using constitutive approaches [10,11,12,13,14,15,16,17,18]. Many models based on rheological principles and mechanical analogs have been developed. Much of the research on residual stresses has focused on technological stresses generated during manufacturing processes, such as polymer curing [31,32,33,34,35], to describe various approaches to modeling and predicting these technological residual stresses. Other works describe experimental methods for assessing technological residual stresses and propose control and optimization strategies [36,37,38,39,40,41]. However, fewer studies have directly investigated viscoelastic behavior and potential residual stress accumulation under cyclic thermomechanical service conditions, particularly relevant for long-term building structure applications under variable temperatures. Works [26,27] address residual stress accumulation in thermoplastic polymers under cyclic thermomechanical effects, and studies [38,42,43] demonstrate stress accumulation under such effects in various composites, including non-polymeric ones.

Thus, operational residual stresses arising under in-service conditions of building structures like industrial gas exhaust ducts and chimneys are significantly less studied than technological residual stresses. Experimental data on residual stress formation and magnitude in thermosetting polymers (including the epoxy polymer considered here) under cyclic thermomechanical effects are insufficient, especially regarding the influence of the initial stress state, comparison of unreinforced polymer and reinforced composite behavior, and residual stress accumulation during holds at constant temperature.

To predict the long-term SSS of polymer composite structures and potential residual stress development and accumulation, predictive models must account for the sequence and duration of thermomechanical effects, capturing the material’s thermomechanical history. Most glass-reinforced plastics in building structures use thermosetting polymer matrices, often epoxy resins (one of which is considered here). Epoxy polymers, even unmodified, exhibit shape memory effects [44], closely related to relaxation transitions, temperature-dependent viscoelastic properties, and stress generation by frozen deformations. Consequently, research on experimental characterization and modeling of thermosetting and thermoplastic polymer viscoelastic behavior is highly relevant [45,46,47,48,49], covering structural models for shape memory polymers, viscoelastic process modeling (creep, relaxation), and transitions through the glass transition temperature. Other approaches, like those using fractional-order derivatives, offer powerful tools for modeling complex hereditary effects. Other approaches, like those using fractional-order derivatives, offer powerful tools for modeling complex hereditary effects [50,51,52,53], although their practical application can be limited by mathematical complexity and parameter identification.

Classical viscoelastic models, such as the Kelvin–Voigt model and its three-element variant [16,18,54,55,56], provide a fundamental basis. The three-element model was chosen for its versatility, allowing consideration of both instantaneous elastic deformation (E1) and subsequent viscoelastic deformation (E2 and *η*), providing a more accurate representation of the studied polymer’s behavior under cyclic thermomechanical loading, including residual stress accumulation. The two-element model, lacking the ability to represent instantaneous elastic deformation, would not adequately capture the material’s initial response to loading, especially during rapid temperature changes. Furthermore, the three-element model offers a good balance between accuracy and computational efficiency, making it suitable for practical engineering applications.

More complex multi-element models, such as the one proposed in [26,27,57], incorporate series-connected Kelvin–Voigt cells with “thermal brakes” to simulate stress accumulation under cyclic thermomechanical conditions considering memory effects. This approach, further developed in our previous work [57], forms the basis of the present study.

This work investigates the stress state of an epoxy polymer, and a glass-reinforced composite based on it under cyclic thermomechanical loading, focusing on residual stress development and magnitude. We combine experimental studies with numerical modeling using an improved version of our previously developed methodology, implemented in Python. The chosen materials, an anhydride-cured epoxy polymer and a glass fabric-reinforced composite, are relevant to chimney and gas duct manufacturing by winding [2].

The objective is to model and explain the observed stress state formation, including potential residual stress accumulation, under varying initial stress levels and heating/cooling cycle parameters. Importantly, this work also aims to obtain new experimental data for the considered epoxy polymer and glass-reinforced composite, which are currently lacking in the literature and essential for practical engineering applications, particularly in industrial gas flues, chimneys, and flue gas treatment systems. Specifically, the influence of the initial stress state and holding at maximum constant temperature on residual stress accumulation under cyclic thermomechanical loading is investigated experimentally for the first time, along with a comparison of unreinforced polymer and reinforced composite behavior.

The main tasks of this work are the following:Develop a Python program for determining the mechanical parameters of the three-element Kelvin–Voigt model from experimental data;Develop a Python program for modeling cyclic thermomechanical loading of pre-loaded clamped rod samples;Determine the temperature-dependent mechanical parameters of the epoxy polymer and glass-reinforced composite;Conduct experimental studies of the stress state of clamped rod samples under cyclic thermomechanical effects, including investigation of the influence of the initial stress state and holds at maximum temperature;Compare the modeling results with experimental data.

## 2. Materials and Methods

### 2.1. Materials

Experimental studies were conducted on rod samples of cured epoxy polymer and glass-reinforced plastic based on it, with reinforcement by glass fabric. The epoxy compound consisted of a hot-curing epoxy resin, an anhydride hardener, and a catalyst to accelerate the curing process. Solvents and plasticizers were not used.

The following materials were used to prepare the epoxy compound:-KER 828 epoxy resin: epoxy group content (EGC) 5308 mmol/kg, equivalent epoxy weight (EEW) 188.5 g/eq, viscosity at 25 °C 12.7 Pa × s, HCl 116 mg/kg, total chlorine 1011 mg/kg. Manufacturer: KUMHO P&B Chemicals, Seoul, Republic of Korea;-Isomethyltetrahydrophthalic anhydride (IZOMTGFA Cheboksary Russia) (hardener for epoxy resin): viscosity at 25 °C 63 Pa × s, anhydride content 42.4%, volatile fraction content 0.55%, free acid 0.1%. Manufacturer: ASAMBLY Chemicals Company Ltd., Nanjing, China.-Alkophen (epoxy curing booster): viscosity at 25 °C 150 Pa × s, molecular formula C15H27N3O, molecular weight 265, amine number 600 mg KOH/g. Manufacturer: JSC “Epital”, Moscow, Russia.

A single composition of the epoxy binder was used with the following mass ratio of components:Epoxy resin (KER 828)—52.5%;Hardener (IZOMTGFA)—44.5%;Curing accelerator (Alkophen)—3%.

The components were mixed using a mechanical disperser-homogenizer Stegler DG-360 (manufacturer: STEGLER, China, Russian representative office: 107076, Moscow, Bogorodskiy val str. 3, info@stegler.ru) with an M-shaped nozzle diameter of 17 mm at a rotation speed of 6000 rpm. After mixing, the binders were poured into silicone molds and placed in a laboratory oven for curing. Samples were cured at 110 °C for 30 min. After primary curing, all samples were held at 150 °C for 12 h to achieve complete polymerization.

After curing and holding, rod samples from unreinforced polymer were cut from the resulting plate, and their surface was additionally leveled using a grinding wheel. The samples were cut in the shape of “dumbbells” (Figure 1a), i.e., with widenings at the ends, which were clamped in the grips of the testing stand. On average, the cross-sectional dimensions of the unreinforced epoxy polymer samples after mechanical processing were about 20–22 × 8.0–8.5 mm, and the distance between the widenings at the ends was 200 mm.

Nuts were glued to the samples for subsequent installation of displacement sensors, which were used to determine the short-term elastic modulus and control deformations during tests (Figure 1a,b). The measurement base of the sensor was from 194 to 196 mm for different samples (tests were conducted on three samples cut from one plate). Three thermocouples were embedded inside one of the samples (top, middle, and bottom) to select the heating and cooling modes of the thermal chamber and control the temperature of the samples during tests (Figure 1c).

Glass-reinforced plastic samples (Figure 1d) were manufactured with a matrix based on the same epoxy binder described above, reinforced with glass fabric EZ-200.

Glass fabric EZ-200 is produced according to GOST 19907-83 [58] and has the following characteristics:Thickness: 0.190 + 0.01/−0.02 mm;Surface density: 200 + 16/−10 g/m^2^;Number of threads per 1 cm of fabric along the warp: 12 ± 1;Number of threads per 1 cm of fabric along the weft: 8 ± 1;Weave: plain;Sizing agent: paraffin emulsion.

Glass fabric sheets EZ-200 were cut and calcined at 300 °C to remove the paraffin sizing agent immediately before impregnation with the binder. A total of 25 layers of glass fabric were used in the samples, laid out according to the scheme 0°/90° (warp/weft).

The mass fraction of glass fiber, determined by weighing the sample fragment before and after the muffle furnace burning of the binder, was 0.43.

Glass-reinforced plastic samples were cured at 110 °C for 30 min in silicone molds with loading through metal plates with a Teflon coating under a pressure of about 0.2 kPa; then, the cured samples were held at 150 °C for 12 h. After that, rectangular (without widenings) rod samples were cut from the plates in the warp direction (0°) considered in this work. On average, the cross-sectional dimensions of the glass-reinforced plastic samples after mechanical processing were about 19.1–19.3 × 4.3–4.5 mm, the length of the samples was 240 mm, and the measurement base of the displacement sensor was from 194 to 195 mm. Thermocouples were glued to the samples on top (and not inside, as in the samples of pure epoxy polymer) due to the small thickness of the samples.

Experimentally (by sequential heating and holding at a constant temperature), the glass transition temperature (transition to the highly elastic state) of the unreinforced polymer was determined; it was 135 °C. The coefficient of thermal expansion, determined by heating the samples in a free state and measuring the elongation using a linear displacement sensor, was 16 × 10^−6^ 1/°C for the unreinforced polymer and 2.4 × 10^−6^ 1/°C for the glass-reinforced plastic.

### 2.2. Methods

#### 2.2.1. Methods of Experimental Research

Tensile tests combined with cyclic thermal effects, as well as the determination of viscoelastic characteristics, were performed on a Tinius Olsen h100ku testing machine (Horsham, PA, USA) within a custom-built thermal chamber.

According to the manufacturer’s specifications, the Tinius Olsen h100ku has a load accuracy of ±0.5% within the range of 0.2–100% of the installed 100 kN force sensor. The crosshead displacement resolution was 0.1 mm, with an error of up to 0.01 mm.

The test setup and procedure are shown in Figure 2. In Figure 2a, the rod sample (1) is clamped within the internal grips (4) of the thermal chamber. The thermal chamber consists of an upper movable part (2) and a lower stationary part (3) rigidly bolted to the stationary base (7) of the Tinius testing machine. The upper part (2) of the thermal chamber is suspended and clamped in the upper grip of the testing machine using a hinged rod (9). The upper (2) and lower (3) parts are separated, allowing the clamped sample (1) within the internal grips (4) to deform freely. A displacement sensor (10) is attached to the sample (1) to measure elongation over a fixed gauge length.

The thermal chamber is internally insulated with mineral wool boards and equipped with heating elements and a fan to ensure uniform heating during heating cycles and cooling during cooling cycles. A thermostat, based on readings from an internal thermocouple, regulates the chamber temperature. Sample temperatures were monitored using thermocouples embedded in control samples.

A control test on steel samples showed that the measurement error due to heating the internal metal parts of the thermal chamber at 100 °C was within ±6 N, approximately 1.5% of the minimum thermal deformation forces of the samples at this temperature (at least 400 N).

The primary cyclic thermomechanical loading experiments were performed on rigidly clamped rod samples of either unreinforced epoxy polymer or glass-reinforced plastic.

Each rod sample was initially clamped in the upper grip and heated to the minimum initial temperature (30 °C in all cases). After reaching this temperature, the sample was clamped in the lower grip, and a tensile load was applied at a rate of 2 mm/min. Upon reaching the target load, the grips were held fixed throughout the experiment, preventing further mechanical deformation. Consequently, the stress in the sample began to decrease due to relaxation. Note that in some tests, no tensile load was applied, and only thermal stresses were induced.

Immediately after applying the load (or initiating the test without mechanical load), heating was initiated. The heating intensity was controlled to maintain approximately the same rate of temperature change during both heating and cooling phases. The precise heating rates and durations for each sample in each test are presented in the Results section.

During heating, the constrained expansion of the sample due to the rigid clamping induced compressive thermal stresses. These stresses are superimposed on the tensile mechanical stresses, reducing the overall stress, as reflected in the stress–time curves. Upon reaching the maximum temperature, the test proceeded in one of two ways, depending on the test protocol: either cooling was initiated immediately or the sample was held at the maximum temperature for a specific duration (5 to 90 min). After the hold period (if applicable), cooling to the minimum temperature was initiated.

During cooling, the sample contracted, generating tensile stresses, including, as shown in the results, accumulating residual stresses. The heating and cooling cycles were repeated the required number of times, with the force readings from the testing machine’s load cell recorded and subsequently converted to stress values. Based on the experimental data, stress–time curves were generated, and temperatures at control points were recorded at 10 °C intervals. The cyclic thermomechanical loading procedure is illustrated in Figure 3.

#### 2.2.2. Methods for Determining the Mechanical Characteristics of the Samples

The approach used in this work and the proposed material model are based on a three-element viscoelastic structural model (Figure 4), sometimes referred to as the three-element Kelvin–Voigt model [16,27], although it may have different names in other sources [18,54,55,56].

The deformation law described by this model is as follows [16]:(1)$$E_{1}\cdot n\cdot \dot{\varepsilon }+H\cdot \varepsilon =m\cdot \dot{\sigma }+\sigma ,$$
where E_1_ is the instantaneous elastic modulus; $H=\frac{E1\cdot E2}{E1+E2}$ is the long-term elastic modulus (where E_2_ is the elastic parameter determined experimentally); m$=\frac{\eta }{E1+E2}$—is the relaxation time; *σ*, $\dot{\sigma }$—are normal stresses and their rate of change; ε, $\dot{\varepsilon }$—are relative strains and their rate of change.

To determine the Kelvin–Voigt model parameters, tensile relaxation tests were performed on samples at different temperatures (30 °C, 60 °C, and 90 °C). The tests were conducted as follows: a constant tensile load was applied to a specimen placed in the thermal chamber and preheated to the required temperature in the unloaded state. The subsequent stress relaxation in the specimen was recorded, and a stress–time curve (relaxation curve) was generated. A displacement sensor was used to measure the initial displacement of the specimen to determine E_1_ (the instantaneous elastic modulus).

To determine the model parameters E_1_, E_2_, *η* Equation (1) is transformed to [16]:(2)$$\left.\sigma (\tau )=H\varepsilon_{0}+(\sigma_{0}-H\varepsilon_{0})exp(\frac{-\tau }{m}\right),$$where *τ* is time, *ε*_0_ = *σ*_0_/E_1_ is the initial strain.

The relaxation curves obtained at 30 °C, 60 °C, and 90 °C, under various initial tensile stresses, for the unreinforced epoxy polymer and GRP are presented in the Results section.

To determine the parameters of the three-element Kelvin–Voigt relaxation model, a Python script was developed to automatically select the E_1_, E_2_, and *η* values that best fit the experimental relaxation curve. The instantaneous elastic modulus, E_1_, is initially determined using the standard method [59], and its value, along with a tolerance of ±0.5%, is set as a constraint. The primary objective is to accurately approximate the experimental stress–time data. The script utilizes the open-source libraries Pandas, NumPy, SciPy, and Matplotlib.

Data are read from an Excel file using Pandas. The core calculations employ the differential evolution method [60], implemented in SciPy, for global optimization. This method effectively finds optimal parameter values by minimizing the error function, even in the presence of multiple local minima in the parameter space. Matplotlib is used for visualization, generating plots of experimental and model data and facilitating the analysis of approximation errors.

The experimental data are divided into user-defined time intervals (any number can be specified). Independent parameter optimization is performed for each interval, allowing the model to capture the material’s behavior at different time scales. The model is based on the relaxation Equation (2). Parameter optimization minimizes the error function, defined as the sum of the squared differences between experimental and model stress values. The resulting parameters for each time interval are then combined to generate a complete model curve. This script effectively adapts the model to changes in material behavior over time, ensuring high approximation accuracy. The results are visualized as graphs, allowing users to assess the agreement between experimental and model data and the optimality of the selected parameters. An example of parameter selection for the three-element viscoelastic model of the unreinforced epoxy binder at 30 °C is shown in Figure 5.

#### 2.2.3. Methods of Theoretical Research

The cyclic thermomechanical loading model described in Section 2.2.1 is based on a multi-element material model. A detailed description and algorithm are presented in our previous work [57]. Here, we provide a summary of the model’s fundamental principles and highlight the key differences and improvements implemented in this study.

The proposed structural model of the polymer material (Figure 6) is multi-element, consisting of *n* serially connected elementary cells (2), each (i-th) of which is a Kelvin–Voigt model characterized by parameters E_1(i)_, E_2(i)_, and *η*_(i)_.

Each cell incorporates a “thermal brake” (1), which conditionally deactivates the cell when it reaches a specific temperature (*t_n_*) and reactivates it upon cooling to the same temperature. Deactivated cells no longer experience stress but continue to undergo virtual deformation according to a defined law (e.g., creep), based on the stress–strain state (SSS) immediately prior to deactivation. These virtual deformations accumulate in the deactivated cells and, upon reactivation, contribute an increment of additional stress, thus modeling stress accumulation and capturing the material’s memory effect. The overall mechanical parameters E_1_, E_2_, and *η*, which describe the material’s rheology at any given time, are determined by summing the parameters E_1(i)_, E_2(i)_, and *η*_(i)_ over all active cells, thus simulating changes in stiffness and rheological properties with temperature.

Under a thermal influence, the unrealized deformation ε_0_ in each temperature increment step induces stresses σ in the elementary cell, which subsequently relax over time. Upon deactivation of a cell at a specific temperature and time (i.e., when the thermal brake is activated), stress redistribution occurs among the remaining active cells, and the deactivated cells undergo virtual deformation according to a defined creep law. The stresses in the deactivated cell become zero, and the total stress is redistributed among the active cells, increasing their load.

When the temperature decreases to the reactivation temperature, the deactivated cells are reactivated. The virtual deformations accumulated during their deactivated state are converted into stresses, which are then summed with the existing mechanical and thermal stresses, distributed evenly among the connected cells. These accumulated stresses, resulting from the virtual deformations, constitute the residual stresses formed due to cyclic thermomechanical loading.

Compared to our previous work, the following improvements were implemented:The algorithm was implemented as a modular Python program (cited in the Appendix A to this paper, referenced at the end of the paper) consisting of a set of functions. This enhances code modularity and flexibility, simplifies maintenance and modification, and facilitates adaptation to other thermomechanical loading conditions in the future;Different laws governing virtual deformation development in deactivated cells were introduced, depending on the stress state at the time of deactivation. Virtual deformations are calculated according to one law if the stress at deactivation is positive and according to a different law if it is negative. This allows for more accurate modeling of material behavior under various loading regimes and temperature effects;The parameters E_2_ and *η* are distributed unevenly among the cells based on the results of their determination from relaxation curves at different temperatures, as described in Section 2.2.2. A parameter distribution function was developed to account for the temperature dependence of material properties, resulting in a more accurate correspondence between the model and experimental data.

The computational algorithm for the cyclic thermomechanical loading model (Figure 3) consists of the following steps:Initialization of model parameters. The initial mechanical characteristics of the material at the initial and final temperatures, the number of deactivated cells within the heating range, sample geometry, mechanical load, and heating/cooling rates (uniform or non-uniform) are defined;Distribution of parameters among cells. Depending on the chosen mode (constant or variable material properties), the parameters E_1_, E_2_, and *η* are distributed among the cells;Calculation of characteristics of each cell. For each cell, the long-term elastic modulus $H=\frac{E1\cdot E2}{E1+E2}$ and the relaxation time parameter *m*$=\frac{\eta }{E1+E2}$ are calculated;Generation of temperature-time points. An array of time points and corresponding temperatures is generated, considering the specified heating and cooling cycles, heating and cooling rates, and any holding periods at constant temperatures;Determination of the number of active cells. At each modeling step, the number of active cells is determined based on the current temperature and the defined temperature at which cells are deactivated;Calculation of total mechanical properties. The parameters E_1_, E_2_, and *η* are summed over the active cells to account for changes in the material’s stiffness and viscoelastic properties during thermal exposure;Calculation of thermal stresses considering relaxation. At each time step, increments of thermal stress are calculated, incorporating relaxation processes using the stress relaxation law (Equation (2));Calculation of mechanical stresses considering relaxation according to law (2) and deactivation of cells. The total mechanical stresses in the active cells are calculated, accounting for their relaxation and stress redistribution upon cell deactivation;Calculation of virtual deformations in disconnected cells. Upon cell deactivation, the accumulation of virtual deformations is modeled, depending on the stress state at the time of deactivation and the duration of the deactivated state. For the considered cyclic thermomechanical loading scheme, with predominant compressive (thermal) stresses prior to cell deactivation, the following law is used:(3)$$\varepsilon_{v}(\tau )=(-\frac{\sigma_{0}}{E_{1}})\cdot (1-exp(-(\tau )\cdot \frac{H}{E_{1}m})),$$

With predominant tensile (mechanical) stresses(4)$$\varepsilon_{v}(\tau )=(-\frac{\sigma_{0}}{E_{1}})\cdot exp(-(\tau )\cdot \frac{H}{E_{1}m}))$$

Accounting for accumulated virtual deformations upon reactivation of cells. During cooling and cell reactivation, the accumulated virtual deformations are converted into additional stresses, which are summed with the current stresses in the material;Summation of total stresses and analysis of results. At each modeling step, mechanical and thermal stresses, along with the additional stresses from virtual deformations, are summed to determine the overall stress–strain state of the material;Visualization and saving of results. The program generates model stress–time curves and saves the calculation results for further analysis and comparison with experimental data.

The program offers flexible configuration of modeling parameters and, with appropriate adjustments, can be extended to loading scenarios beyond those considered in this study.

Figure 7 and Figure 8 illustrate how the developed model explains the experimental data. These figures depict key stages of the model algorithm and the mechanisms used to describe observed effects, such as residual stress accumulation and its dependence on initial conditions. The aim is not only to describe the approach but also to demonstrate its application in explaining specific experimental results, visually connecting the theoretical algorithm with practical observations. Therefore, the following information is best understood in conjunction with the experimental results presented in Section 3.2.

Figure 7 shows the initial stage (a), where no mechanical or thermal stresses are applied. The numerical designations in Figure 7 are

Conditional line representing the neutral position of cells;Cell in the neutral position;Conditional thermal brake in the “on” state.

Figure 8 illustrates the heating and cooling stages, including cell deactivation and reactivation, stress redistribution, and the development of virtual deformations in the deactivated state. Figure 8 shows

Stage (c): Application of mechanical tensile load;Stages (d), (e), (f): Heating;Stages (g), (h), (i): Cooling;Stage (j): Return to the initial temperature.

The numerical designations in Figure 8:4.Conditional line denoting the stretched position of cells under mechanical load;5.Connected cell with tensile (positive) stresses;6.Disconnected cell upon reaching a certain temperature t_n_, in which there were tensile stresses before disconnection;7.Conditional thermal brake in the off state;8.Disconnected cell upon reaching a certain temperature t_n_, in which there were compressive stresses before disconnection;9.Cell reconnected during cooling with reduced positive stresses redistributed in the disconnected state;10.Cell reconnected during cooling with accumulated positive stresses.

The following is a brief explanation of Figure 7 and Figure 8. After the initial stage, mechanical tensile stress is applied at stage (I), causing all cells to deform to position 4. At stage (II), as the temperature increases, the cell initially under mechanical tensile stress is deactivated. This freed cell begins to undergo virtual deformation (creep) toward the neutral position (i.e., it effectively shortens). Concurrently, the tensile stresses in the connected cells decrease due to relaxation and the increasing compressive thermal stresses. At stage (III), the next cell is deactivated; it still experiences tensile stresses before deactivation. At stage (IV), the next cell is deactivated; it experiences compressive stresses before deactivation, resulting in virtual deformation that increases its length. A similar process occurs at stages (V). At stage (VI), as the temperature decreases, cell reactivation occurs; the accumulated virtual deformations are manifested as additional tensile stresses (because the effective cell length increased during virtual deformation). Consequently, the compressive stresses redistributed from the connected cells to the reactivating cell are partially compensated. At stages (VI, VIII), the next cells are reactivated, also receiving an increment of tensile stress. In stage (IX), all cells are reactivated and have accumulated residual tensile stresses.

This process models the material’s memory of thermomechanical loading. Virtual deformations accumulate in the cells, and their accumulation rate and magnitude depend on previous states. Under certain combinations of initial stress state, heating/cooling cycle parameters, and viscoelastic parameters, residual stress accumulation occurs, consistent with both modeling and experimental results. However, under other parameter combinations, accumulation may not occur or may have the opposite sign.

## 3. Results

### 3.1. Relaxation Studies in Unreinforced Polymer and Glass-Reinforced Plastic

Relaxation tests of mechanical stresses under tension were conducted on rod samples of unreinforced epoxy polymer and glass-reinforced plastic based on it, according to the methodology described in Section 2.2.2.

The resulting relaxation curves for the unreinforced epoxy polymer at 30 °C, 60 °C, and 90 °C, under various initial tensile stresses, are shown in Figure 9. The relaxation curves for the glass-reinforced plastic are shown in Figure 10.

Table 1 presents the approximated parameters E1, E2, and *η* for the three-element viscoelastic model for unreinforced epoxy polymer, Table 2 presents the approximated parameters E1, E2, and *η* for GRP. These parameters were calculated using the developed Python script, based on the experimental relaxation curves. The parameters were determined for different time intervals, with the number and range of these intervals chosen to optimize the fit between the model’s relaxation curve (based on the selected parameters) and the experimental relaxation curve. These derived parameters were then selectively used for modeling the primary cyclic thermomechanical loading experiments. In Table 1, the values that will be used in the simulation are highlighted in bold and color (reference is made below when describing the simulation results).

### 3.2. Experimental Studies of Unreinforced Epoxy Polymer Under Cyclic Thermomechanical Loading

The results of experimental studies on the unreinforced epoxy polymer under cyclic thermomechanical loading, as described in the Materials and Methods section, are presented in Figure 11, Figure 12, Figure 13, Figure 14, Figure 15 and Figure 16. The results are shown as stress–time curves for the rod samples during heating, cooling, or holding at a constant temperature. Data points on the curves correspond to recorded stress and time values at 10 °C temperature intervals (in some cases, 5 °C).

The descending portions of the curves represent the heating process and the corresponding decrease in tensile stress; conversely, the ascending portions represent cooling and the increase in tensile stress, including residual stress accumulation (where observed).

Tests were conducted under the following varying conditions:Different maximum heating temperatures (80 °C and 90 °C);Different holding times at the maximum temperature (5 to 90 min);Different initial tensile stresses (0 to 4.3 MPa).

The minimum cooling temperature was consistently 30 °C. The holding time at the maximum temperature also varied within some experiments from cycle to cycle; for example, it could be 5 min in the first cycle, 15 min in the second, 45 min in the third, and 90 min in the fourth (specific data are provided in the figure captions).

Figure 11, Figure 12 and Figure 13 show the results of experiments with heating to 90 °C, with and without holds, at initial stresses ranging from 1.5 to 4.3 MPa. A stress of 1.5 MPa does not exceed the absolute value of the induced thermal stresses, which can be estimated from Figure 14, Figure 15 and Figure 16; stresses of 4 MPa approximately twice exceed thermal stresses.

The results in Figure 11 show that the curves for each cycle are nearly identical; the stresses at the lower temperature boundary (30 °C during cooling) align and do not exhibit a significant tendency to form or accumulate residual tensile stresses. The results in Figure 12 also show no clear tendency for residual tensile stress accumulation; instead, a slight downward trend of the line connecting the upper extrema (stresses at 30 °C) is observed. This could be attributed to the continued relaxation of the initial tensile stresses or the possible accumulation of small residual compressive stresses.

In Figure 13, the tensile stresses after the first and subsequent cooling cycles clearly exceed the initial mechanical tensile stress. The graph shows the formation and accumulation of residual tensile stresses during cooling, with gradual damping and stabilization. The maximum tensile stresses at the end of the experiment are approximately 1.7 times greater than the initial stresses.

These experimental results indicate that residual stress formation in the epoxy polymer depends on the initial stress state. At high initial tensile stresses, residual stress formation and accumulation are suppressed. Conversely, with decreasing initial mechanical tensile stress, a distinct appearance and accumulation of residual tensile stresses is observed. The trend shown in Figure 14 is similar to that in Figure 13, exhibiting a similar tendency for residual tensile stress formation and accumulation, albeit with different absolute values due to variations in initial stress and maximum temperature. At the end of the experiment, the tensile stresses were approximately 2.1 times greater than the initial stresses. In relative terms, the effect is more pronounced despite a lower maximum temperature (80 °C compared to 90 °C), likely due to the lower initial tensile stress.

Figure 15 and Figure 16 show the results of experiments that differed only in the initial tensile stress level. In Figure 15, the initial stress was zero (i.e., no mechanical tension was applied, and all stresses were thermal). In Figure 16, the initial stress was 1.1 MPa. Both experiments included holds at a maximum temperature of 80 °C.

Both figures clearly show the formation and accumulation of residual tensile stresses, with a tendency toward damping and stabilization at the end of the experiment. With zero initial mechanical stress (Figure 15), the residual stresses reached 1.9 MPa. While the relative magnitude of the effect cannot be directly evaluated due to the zero initial stress, the absolute magnitude demonstrates a significant accumulation of residual stress.

Comparing Figure 16 (with holds) and Figure 14 (without holds), which show results for comparable initial mechanical stresses, reveals a noticeable difference in residual stress accumulation. With holds, the final tensile stresses were 2.7 times greater than the initial stresses, whereas, without holds, this increase was only 2.1 times. This suggests that relaxation processes occurring during the holds at a constant temperature subsequently contribute to the increase in residual stresses during cooling.

The experimental results presented in this section demonstrate that residual stress formation in the epoxy polymer is significantly influenced by the initial tensile stress level. At high initial stresses, residual tensile stress accumulation is not observed, likely due to a balance between thermal and mechanical effects. Conversely, at low initial stresses or in the absence of initial mechanical stress, residual tensile stress accumulation is pronounced. This effect is further enhanced by holds at the maximum temperature, indicating the influence of relaxation processes occurring at elevated temperatures. The final residual stresses increase with increasing hold time and, in relative terms, are more significant with decreasing initial mechanical stress.

### 3.3. Results of the Experimental Studies of Glass-Reinforced Plastic Under Cyclic Thermomechanical Loading

The results of experimental studies on the glass-reinforced plastic under cyclic thermomechanical loading, as described in the Materials and Methods section, are presented in Figure 17, Figure 18 and Figure 19. 

Residual stress accumulation in the glass-reinforced plastic is significantly less pronounced compared to the pure polymer, even though the matrix of the studied composite is the same epoxy binder. This is likely due to the reinforcing effect of the glass fabric, which reduces thermal expansion and provides higher residual stiffness at elevated temperatures, thereby mitigating the influence of thermal cycles. However, the stress–time curves under cyclic loading still show a slight increase in tensile stress during cooling with increasing cycles, although it is less pronounced than in the pure polymer. This is particularly evident in Figure 17, where the residual stress accumulation in the upper part of the curve, at zero initial stress, resembles the behavior of the pure polymer, albeit with significantly lower absolute stress values.

In Figure 18, an initial decrease in stress is observed, likely due to the relaxation of the initial mechanical stress. Subsequently, a slight increase in stress can be seen, particularly with longer hold durations. Figure 19 also shows a minimal increase in residual stress; however, this increase is close to the measurement error, making accurate interpretation challenging.

Future work will focus more closely on the study of glass-reinforced plastics, as their properties, including the influence of reinforcement, warrant further investigation. Particular attention will be paid to the material’s long-term behavior, as effects related to residual stress accumulation may not be immediately apparent but may manifest after extended periods. This is especially important considering that glass-reinforced plastic structures in the construction industry are designed for decades of service. A thorough understanding of these processes will enable more accurate predictions of composite material durability under real operating conditions.

As demonstrated in Section 3.2, significant residual stresses were observed in the unreinforced polymer under cyclic heating between 80 °C and 90 °C. In contrast, no significant residual stresses were observed in the GRP at these temperatures. Consequently, the temperature was progressively increased for the GRP experiments, up to the upper limit for long-term application, 180 °C. At this elevated temperature, some stress accumulation was observed in the GRP, but it was considerably less pronounced than that observed in the unreinforced polymer at lower temperatures.

### 3.4. Modeling Cyclic Thermomechanical Loading on Unreinforced Epoxy Polymer

In this work, the glass-reinforced plastic (GRP) was studied less extensively than the unreinforced epoxy polymer, which serves as its matrix. This focus on the pure polymer was intentional, prioritizing its characterization for subsequent modeling.

The modeling presented in this article serves to illustrate the proposed approach rather than provide a comprehensive analysis of all experimental cases. Several representative experiments were selected to demonstrate the model’s ability to analyze cyclic thermomechanical loading and describe residual stress accumulation in the epoxy polymer.

The mechanical parameters used for modeling are presented in Table 1. They were selected for time intervals approximating the duration of one heating/cooling cycle to align with experimental conditions. However, a more rigorous analysis may require considering parameters for longer time intervals or refining the selection method, which warrants further investigation.

Figure 20 shows the modeling results for the experiment described in Figure 15: heating to 80 °C with holds and zero initial tensile stress (i.e., only thermal stresses). The parameters used for this simulation are highlighted in green bold text in Table 1.

Figure 20a shows the model curves generated using Matplotlib, as follows:The blue curve represents the stress without accounting for additional stresses from virtual deformations;The red curve represents the stress accounting for stresses from virtual deformations.

Experimental data are shown in Figure 20b for comparison.

The model curves closely match the experimental data, both with and without considering virtual deformation stresses. However, the absolute tensile stress values at the lower temperature boundary (during cooling) are significantly closer to the experimental data when virtual deformation stresses are included. The experimental maximum tensile stress was 1.88 MPa, while the model predicted approximately 1.0 MPa without considering additional stresses (a difference of approximately 90%) and approximately 1.55 MPa with their inclusion (a difference of approximately 20%).

The numerical stress values at the upper temperature boundary (80 °C) are nearly identical between the model and experimental curves, both in trend and magnitude. This confirms the validity of the parameter selection and generally supports the adequacy of the proposed approach.

It is noteworthy that the cooling curves exhibit a clear nonlinearity, while the heating curves are nearly linear. This trend is observed in both the experimental and model data, even though the model parameters were derived from relaxation curves obtained under different conditions (without thermal stresses). This agreement between experimental and calculated curve shapes suggests that the experimental data are reliable and not significantly distorted by unforeseen factors (e.g., equipment artifacts).

Figure 21 shows the modeling results for the experiment described in Figure 11: heating to 90 °C without holds and with an initial tensile stress of 4.3 MPa. The parameters used for this simulation are highlighted in red bold text in Table 1.

Figure 21a shows the model curve for thermal stresses, Figure 21b shows the model curve for mechanical tensile stresses, and Figure 21c shows the total stress curves (blue—without considering additional residual stresses, red—with consideration). Experimental data are shown in Figure 22 for comparison.

In this case, the shape and trend of the experimental curve best match the model curve representing only mechanical stresses (Figure 21b). The absolute stress values at the lower temperature boundary (30 °C) are almost identical, while at the upper temperature boundary, the purely mechanical stresses (as expected) differ from the experimental data by the magnitude of the thermal stresses.

The primary conclusion is that under the experimental conditions considered, residual thermal stress accumulation is not observed, likely due to the initial stress state of the samples—specifically, a significant excess of tensile mechanical stress over thermal stress.

However, if modeling is performed without considering the additional stresses arising from virtual deformations, the resulting model curve differs significantly from the experimental data (blue curve in Figure 21c). By incorporating the additional stresses from virtual deformations, as proposed in this work, a qualitative improvement in the model’s agreement with the experimental data is achieved; the peak stress values are reduced (red curve in Figure 21c).

Thus, this latter modeling approach partially captures the main experimental trends. However, despite the improved agreement in curve shape, the quantitative values of the residual stresses do not yet fully match the experimental data, indicating the need for further refinement of the approach and model parameters.

The selection of appropriate material mechanical characteristics remains an important consideration. Further attention should also be given to the law governing virtual deformation development in deactivated cells. In this modeling effort, this law was chosen somewhat empirically, although it does allow for some approximation of the calculated values to the experimental ones. While the proposed approach represents a positive step toward accurately describing the residual stress accumulation process, several unresolved issues remain.

The schemes presented below in Figure 7 and Figure 8 illustrate an attempt to explain the obtained experimental data using the developed model. These schemes reflect the key stages of the model algorithm and mechanisms laid down to describe the observed effects, such as the accumulation of residual stresses and their dependence on initial conditions.

In Figure 7, the initial stage (a) is shown, where no mechanical or thermal stresses are applied to the sample. Numerical designations in Figure 7 are as follows:

1—conditional line of the neutral position of cells;

2—cell in neutral position;

3—conditional thermal brake in the on state.

Figure 8 shows the stages of heating and cooling, accompanied by disconnection and connection of cells, stress redistribution between cells, and the development of virtual deformations in the disconnected state.

In Figure 22, the following are presented:Stage (c)—application of mechanical tensile load;Stages (d), (e), (f)—heating;Stages (g), (h), (i)—cooling.Stage (j)—return to the initial temperature.

Numerical designations in Figure 22 are as follows:

4—conditional line denoting the stretched position of cells under mechanical load;

5—connected cell with tensile (positive) stresses;

6—disconnected cell upon reaching a certain temperature t_n_, in which there were tensile stresses before disconnection;

7—conditional thermal brake in the off state;

8—disconnected cell upon reaching a certain temperature t_n_, in which there were compressive stresses before disconnection;

9—cell reconnected during cooling with reduced positive stresses redistributed in the disconnected state;

10—cell reconnected during cooling with accumulated positive stresses.

We will provide a brief description to explain the schemes in Figure 7 and Figure 8.

After the initial stage (a), at stage (b), a mechanical tensile stress is applied to the sample; all cells deform to position 4.

At stage (c), the temperature increases, and the moment of cell disconnection is shown, which was loaded with mechanical tensile stresses. The freed cell begins to virtually deform according to the creep law toward the neutral position (i.e., conditionally shortens its length). At the same time, in the connected cells, tensile stresses decrease due to relaxation and the imposition of compressive thermal stresses that begin to develop with increasing temperature.

At stage (d), the next cell is disconnected; in it, tensile stresses still prevailed before disconnection.

At stage (e), the next cell is disconnected; before disconnection, compressive stresses already prevail in it; as a result, after release, the cell begins to increase its length (virtual deformations develop according to another creep law). Similarly, this occurs at stage (f) with the next cell.

At stage (g), the temperature decreases, and the moment of cell reconnection is shown; accumulated virtual deformations are frozen in it, which gives additional tensile stresses (since the conditional length of the cell increased during the time of virtual deformation). Accordingly, the compressive stresses that are redistributed from the connected cells to the reconnecting cell at the moment of reconnection are somewhat compensated. At stage (h), the next cell is reconnected; it also receives an increment of tensile stresses.

Thus, the memory effect on thermomechanical impact is modeled, i.e., virtual deformations accumulate in the cells, and the accumulation law and their magnitude depend on previous states. Under certain ratios of the initial stress state, parameters of heating and cooling cycles, and values of viscoelastic parameters, residual stress accumulation occurs, which was obtained both from the modeling results and from experiments. At the same time, under other parameter ratios, accumulation may not occur or may have the opposite sign.

## 4. Discussion

This work presented experimental and theoretical studies of the viscoelastic behavior of an epoxy polymer and GRP based on it under cyclic thermomechanical loading. The primary goal was to investigate residual stress formation and accumulation as a function of initial mechanical stress and temperature cycle parameters.

The experimental results showed significant residual tensile stress accumulation in the pure epoxy polymer at low initial mechanical stresses or in their absence. This effect was enhanced by holds at the maximum temperature, indicating the influence of relaxation processes at elevated temperatures. At high initial tensile stresses, residual stress accumulation was not observed, likely due to a balance between mechanical and thermal stresses, as well as the specific viscoelastic properties of the material at different loading levels.

In the GRP, residual stress accumulation was significantly less pronounced. This is likely attributable to the reinforcing effect of the glass fabric, which increases material stiffness and reduces the coefficient of thermal expansion, thus mitigating the influence of thermal cycles. Nevertheless, a slight increase in residual stress was still observed, especially during extended holds at the maximum heating temperature.

Modeling using an enhanced multi-element model based on the three-element Kelvin–Voigt model demonstrated qualitative and quantitative agreement with the experimental data for the epoxy polymer. By incorporating virtual deformations in deactivated cells at specific temperatures, the model accounts for the material’s memory of thermomechanical loading. This approach provided insights into the mechanism of residual stress accumulation and its dependence on initial conditions and heating/cooling cycle parameters.

A comparison of experimental and calculated data confirmed the general adequacy of the proposed approach. However, some discrepancies in the quantitative values of residual stresses highlight the need for further model refinement. More accurate determination of the material’s mechanical parameters at different temperatures and relaxation times is required, as is further investigation into the laws governing virtual deformation development as a function of the stress state of cells prior to deactivation.

These findings are consistent with previous studies highlighting the complex nature of viscoelastic behavior in polymers and composites under cyclic thermomechanical loads [26,27,42,43,61]. Residual stress accumulation can significantly affect the durability and performance of polymer structures, particularly under long-term operation at variable temperatures.

The key qualitative and quantitative results of this study are

Significant residual tensile stress accumulation is observed in the epoxy polymer at low initial mechanical stresses (0 to 1.5 MPa);Maximum residual stresses exceed the initial stresses by factors of 1.7–2.7, reaching values up to 1.9 MPa at zero initial stress and up to 2.1 MPa at an initial stress of 0.8 MPa;Holds at the maximum temperature (up to 90 min) enhance residual stress accumulation;At high initial tensile stresses (approximately 4 MPa), exceeding thermal stresses, residual stress accumulation is not observed, likely due to relaxation at elevated temperatures;Residual stress accumulation is significantly weaker in the GRP. Maximum residual stresses in the GRP did not exceed 0.2–0.3 MPa, less than 10% of the initial stresses. The glass fabric reinforcement increases stiffness and reduces thermal expansion, mitigating the influence of thermal cycles;The developed multi-element Kelvin–Voigt model adequately describes the experimental data for the epoxy polymer, accounting for the memory effect and predicting residual stress accumulation as a function of initial conditions and cycle parameters.

Future research will focus on expanding the experimental database for GRPs with various reinforcement types and matrices, as well as further refining the model to incorporate longer relaxation processes. This will improve the accuracy of predicting the stress–strain state and residual stresses in polymer composites, which is crucial for ensuring structural reliability and durability.

The model could also be extended to account for transitions to a highly elastic state, including the forced elastic state (Figure 23). This could be achieved, for instance, by adding a parallel elastic element with an elastic modulus (E_9_) corresponding to the glass transition temperature, which would remain active after all other cells are deactivated. To model the forced elastic state transition, cell deactivation criteria could be based not only on temperature, as in the current model, but also on the stress level (σ_1_ − σ_n_) within the cell. This will be the subject of future research.

## 5. Conclusions

This study investigated residual stress accumulation in an epoxy polymer and a glass-reinforced composite under cyclic thermomechanical loading, considering the influence of initial stress levels, temperature profiles (including holds), and reinforcement. The key findings are summarized below, as follows:The initial stress state significantly governs residual stress accumulation in the pure epoxy polymer. Low initial tensile stresses (0–1.5 MPa) resulted in substantial residual stress accumulation, exceeding initial stresses by up to 2.7 times and reaching values up to 2.1 MPa. Conversely, high initial stresses (around 4 MPa) suppressed residual stress accumulation due to the dominance of relaxation processes. This highlights the critical role of initial loading conditions in predicting long-term material behavior. We consider this result to be one of the most interesting and important, since there is no literature information that the residual stress formation in unreinforced epoxy polymer can be completely suppressed at high initial mechanical stress level;Reinforcement with glass fibers effectively mitigates residual stress accumulation in the composite. The glass-reinforced plastic exhibited significantly lower residual stresses (less than 10% of initial stresses) compared to the neat polymer, demonstrating the beneficial effect of reinforcement in reducing the impact of thermal cycling;The developed multi-element model effectively captures the material’s “memory” of thermomechanical history, enabling reasonable prediction of residual stress accumulation. The model, based on the three-element Kelvin–Voigt model, demonstrated satisfactory agreement with experimental data for the epoxy polymer, capturing the key trends observed in the experiments. This model provides a valuable tool for predicting the stress–strain state of polymeric materials under complex thermomechanical loading and offers a basis for future refinements, including the incorporation of a forced elastic state and more detailed consideration of material parameters and virtual deformation laws. Further research will focus on expanding the experimental database for various composite materials and refining the model to improve prediction accuracy for long-term structural applications.

## Figures and Tables

**Figure 1 polymers-17-00344-f001:**
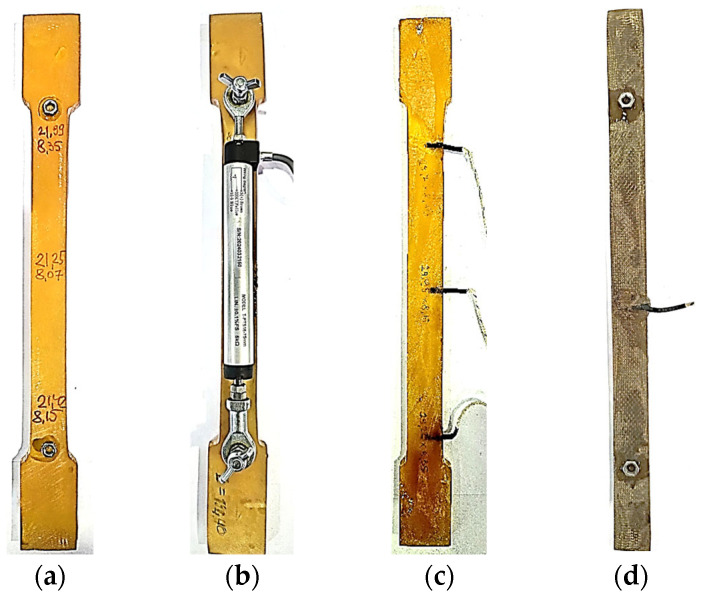
Experimental samples of cured epoxy polymer: (**a**) experimental sample of unreinforced epoxy polymer; (**b**) experimental sample of polymer with installed displacement sensor; (**c**) experimental sample of polymer with embedded thermocouples; (**d**) experimental sample of glass-reinforced plastic with glued thermocouple.

**Figure 2 polymers-17-00344-f002:**
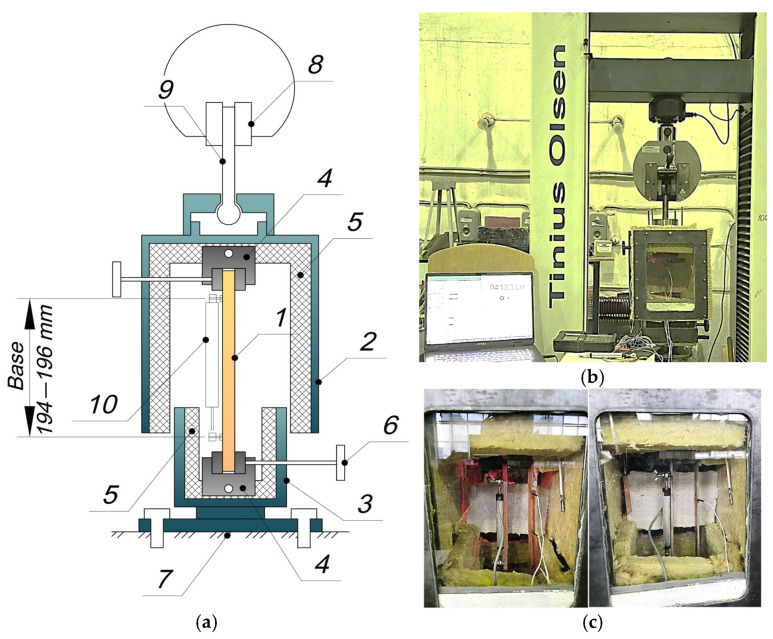
Testing setup: (**a**) diagram; (**b**) general view; (**c**) testing process of polymer (**left**), glass-reinforced plastic (**right**): 1—the rod sample, 2—the upper part of the thermal chamber, 3—lower stationary part of the thermal chamber, 4—internal grips, 5—thermal insulation, 6—gripper handle, 7—stationary base of the Tinius testing machine, 8—top gripper of the Tinius testing machine, 9—hinged rod, 10—displacement sensor.

**Figure 3 polymers-17-00344-f003:**
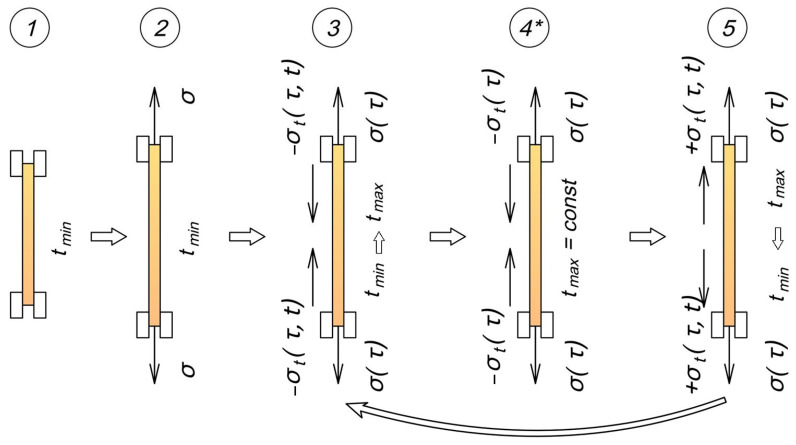
Schematic of experiment stages (numbered) under cyclic thermomechanical loading: 1—initial state at temperature t_min_, 2—tensile stresses σ are applied, temperature t_min_, 3—temperature increases from t_min_ to t_max_, mechanical tensile stresses σ relax with time τ, compressive thermal stresses σ_t_ increase in modulus, 4—stage of holding at constant temperature t_max_ (*—means that this stage is present only in those experiments where a constant maximum temperature hold is provided), 5—temperature decreases from t_max_ to t_min_, mechanical tensile stresses σ relax with time τ, temperature stresses σ_t_ become tensile and increase in modulus. Thin arrows show the direction of stress action (tension or compression), contour arrows indicate transition processes between stages.

**Figure 4 polymers-17-00344-f004:**
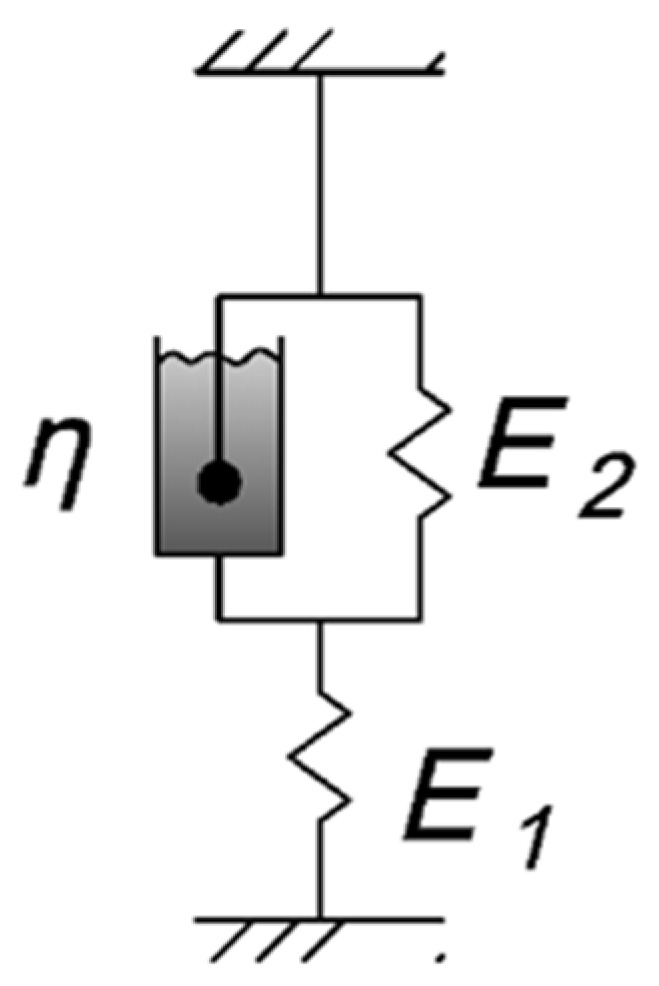
Scheme of the viscoelastic three-element Kelvin–Voigt model. (E_1_, E_2_—elastic parameters, *η*—viscosity).

**Figure 5 polymers-17-00344-f005:**
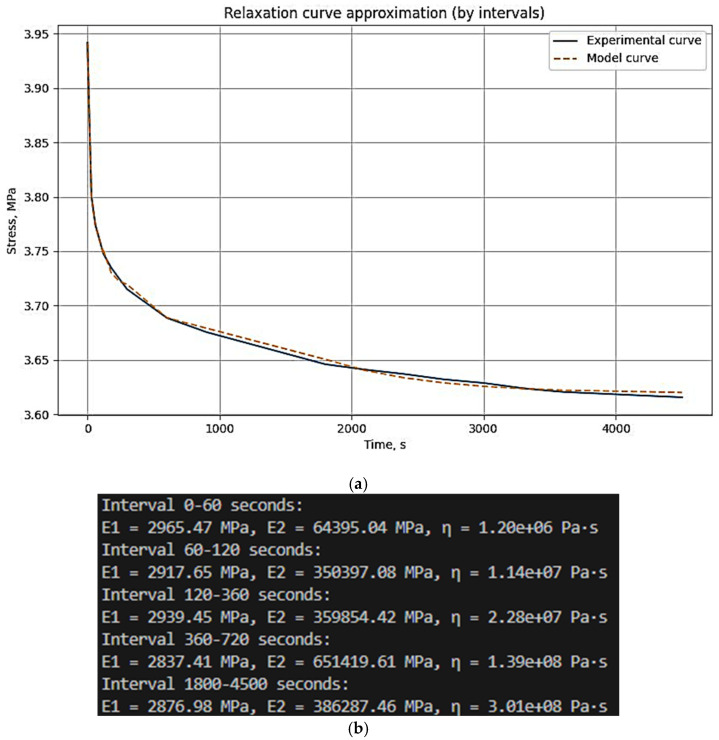
Example of parameter selection for the three-element viscoelastic model on the example of relaxation of unreinforced epoxy polymer at 30 °C and initial stresses of 3.95 MPa: (**a**) experimental and model curve; (**b**) parameter selection results (screenshot from the program code) E_1_, E_2_, *η* when divided into time intervals (0–60 s, 60–120 s, 120–360 s, 360–720 s, 1800–4500 s).

**Figure 6 polymers-17-00344-f006:**
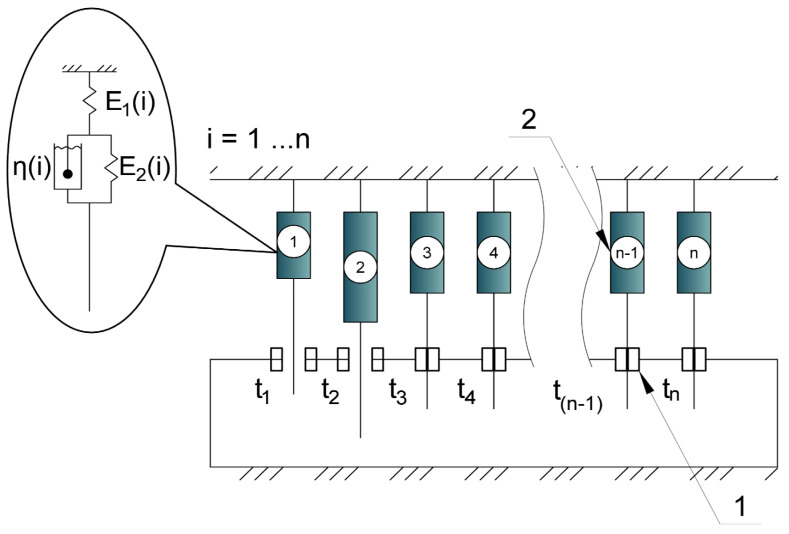
Model scheme: 1—thermal brake; 2—cell in the form of a three-element viscoelastic model.

**Figure 7 polymers-17-00344-f007:**
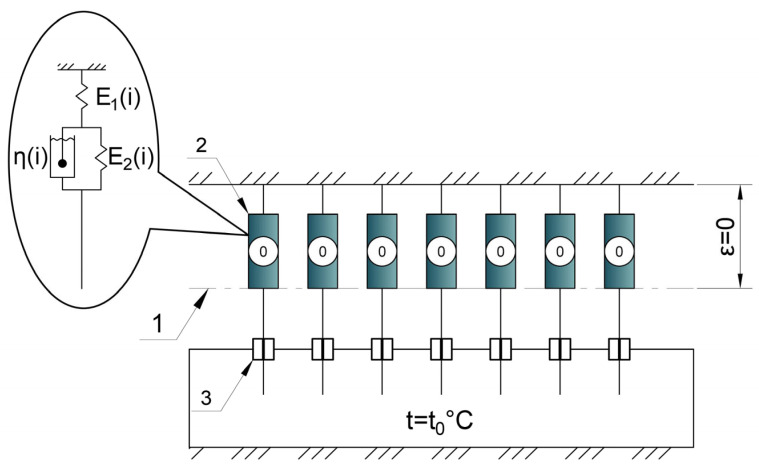
Schematic description of the algorithm of the proposed model at the initial stage (a): no applied loads and initial temperature: 1—conditional line representing the neutral position of cells, 2—cell in the neutral position, 3—conditional thermal brake in the “on” state.

**Figure 8 polymers-17-00344-f008:**
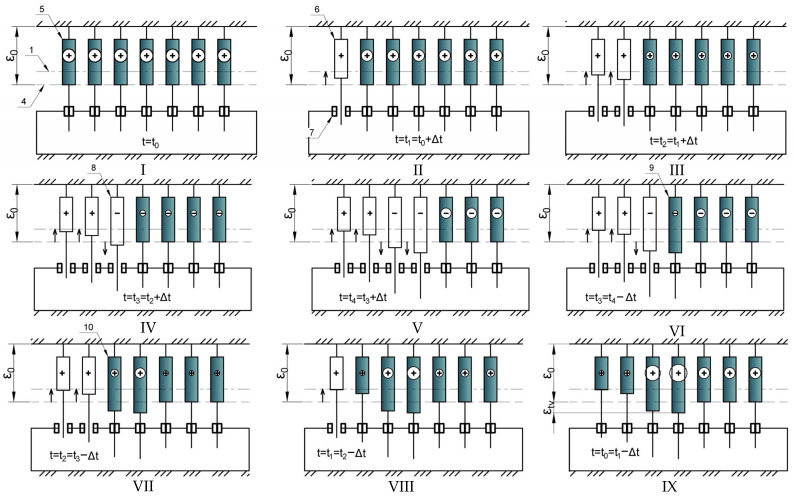
Schematic description of the algorithm of the proposed model.: 1—conditional line representing the neutral position of cells, 4—conditional line denoting the stretched position of cells under mechanical load, 5—connected cell with tensile (positive) stresses, 6—Disconnected cell upon reaching a certain temperature t_n_, in which there were tensile stresses before disconnection, 7—conditional thermal brake in the off state, 8—Disconnected cell upon reaching a certain temperature t_n_, in which there were compressive stresses before disconnection, 9—Cell reconnected during cooling with reduced positive stresses redistributed in the disconnected state, 10—Cell reconnected during cooling with accumulated positive stresses. Roman numerals indicate the stages of the modeled cyclic impact, their brief description is given above in the text.

**Figure 9 polymers-17-00344-f009:**
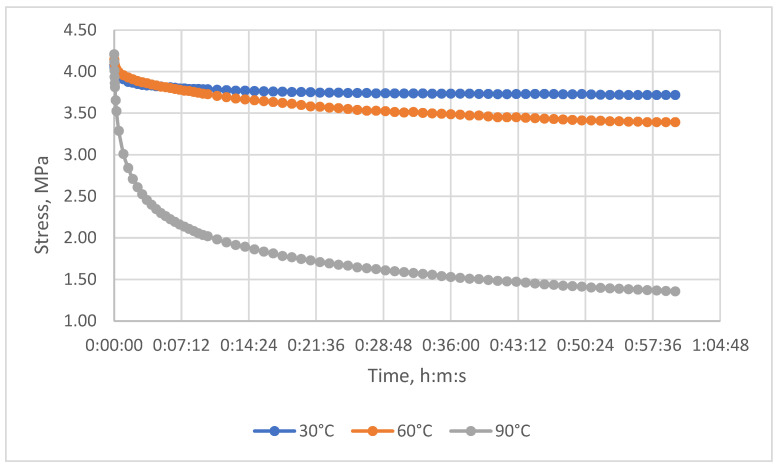
Relaxation curves of the investigated epoxy polymer at temperatures of 30 °C, 60 °C, and 90 °C and initial tensile stresses of 4.2 MPa.

**Figure 10 polymers-17-00344-f010:**
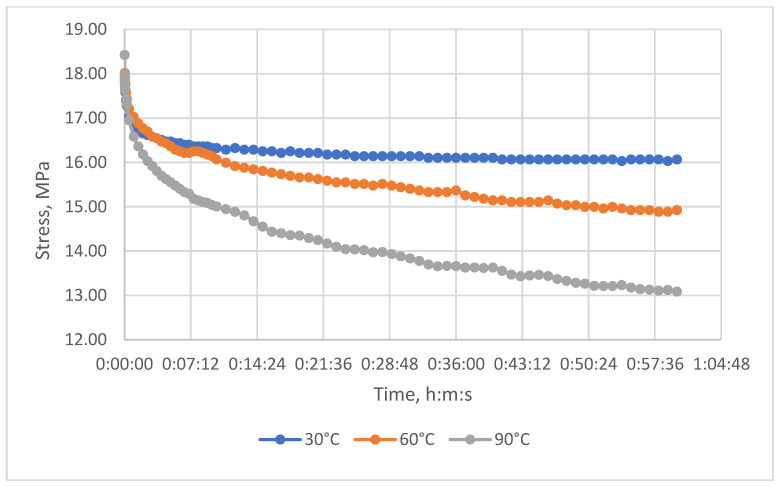
Relaxation curves of the investigated glass-reinforced plastic at temperatures of 30 °C, 60 °C, and 90 °C and initial tensile stresses of 18.4 MPa.

**Figure 11 polymers-17-00344-f011:**
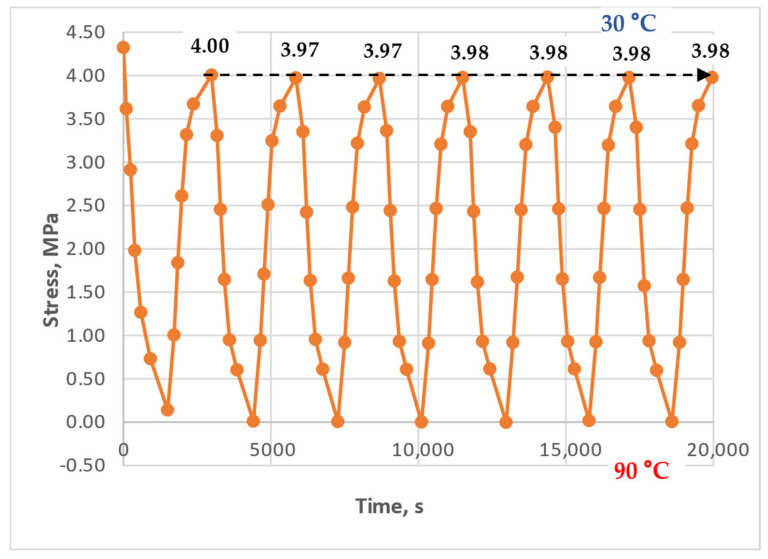
Dependence of stresses in the polymer sample on time under cyclic mechanical loading (initial stress 4.3 MPa, heating to 90 °C, without holds).

**Figure 12 polymers-17-00344-f012:**
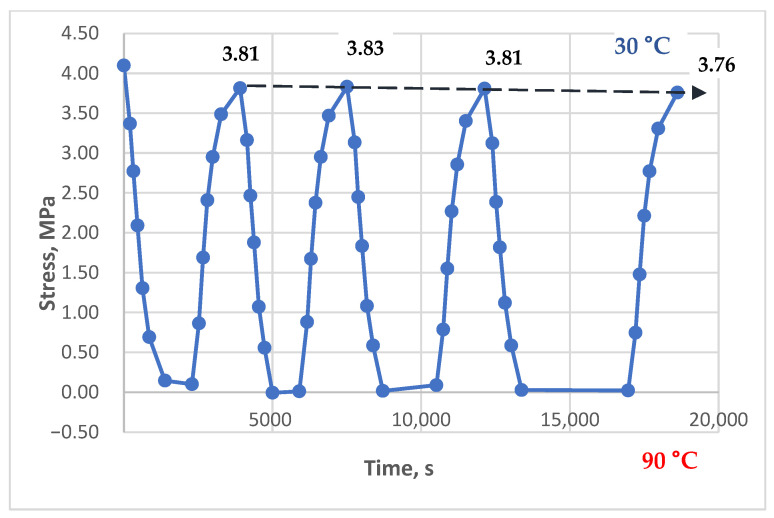
Dependence of stresses in the polymer sample on time under cyclic mechanical loading (initial stress 4.1 MPa, heating to 90 °C, holds at 90 °C—15 min, 15 min, 30 min, 60 min).

**Figure 13 polymers-17-00344-f013:**
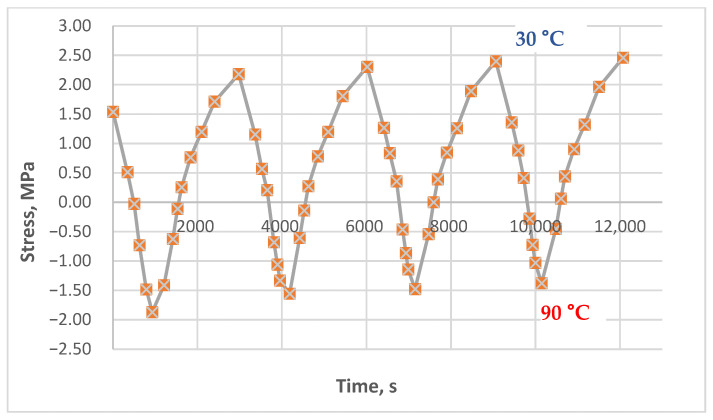
Dependence of stresses in the polymer sample on time under cyclic mechanical loading (initial stress 1.5 MPa, heating to 90 °C, without holds).

**Figure 14 polymers-17-00344-f014:**
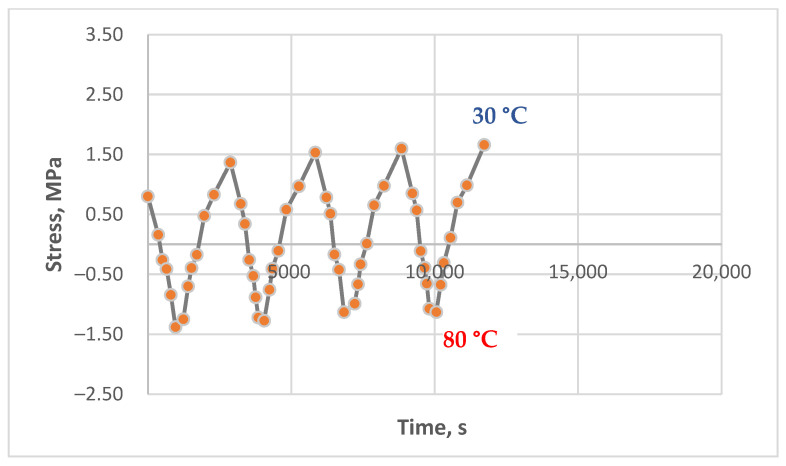
Dependence of stresses in the polymer sample on time under cyclic mechanical loading (initial stress 0.8 MPa, heating to 80 °C, without holds).

**Figure 15 polymers-17-00344-f015:**
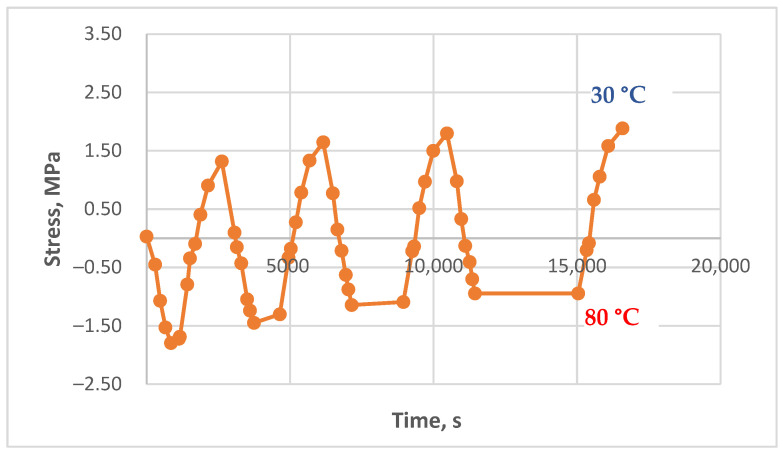
Dependence of stresses in the polymer sample on time under cyclic mechanical loading (initial stress 0.0 MPa, heating to 80 °C, holds at 80 °C—5 min, 15 min, 45 min, 90 min).

**Figure 16 polymers-17-00344-f016:**
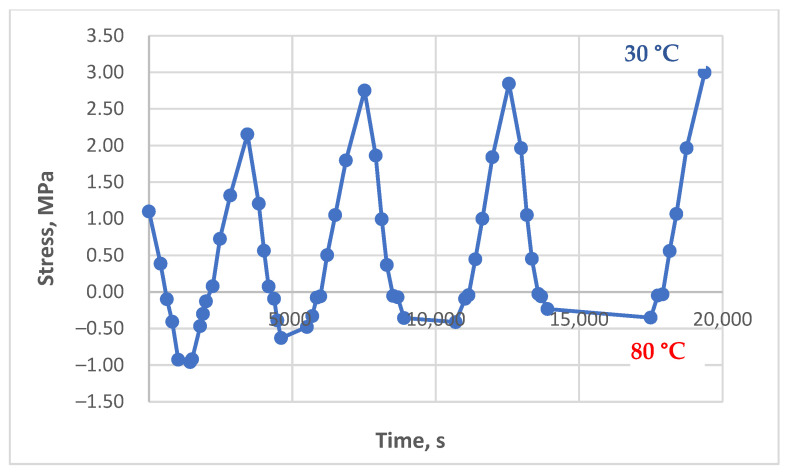
Dependence of stresses in the polymer sample on time under cyclic mechanical loading (initial stress 1.1 MPa, heating to 80 °C, holds at 80 °C—5 min, 15 min, 45 min, 90 min).

**Figure 17 polymers-17-00344-f017:**
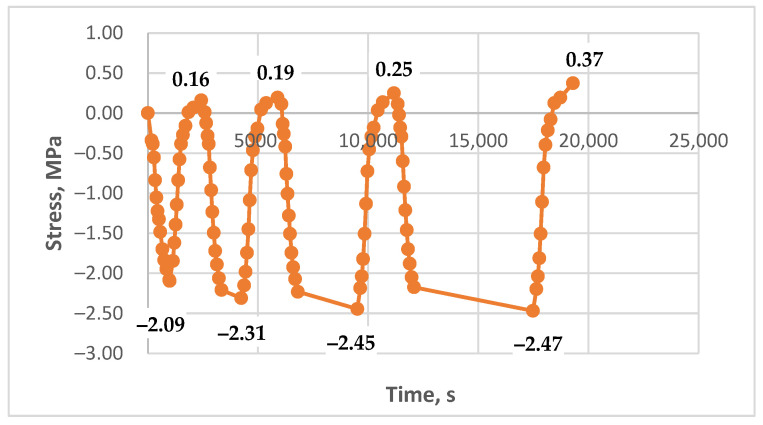
Dependence of stresses in the glass-reinforced plastic sample on time under cyclic thermomechanical loading (initial stress 0.0 MPa, heating to 150 °C, holds at 150 °C—1 min, 15 min, 45 min, 90 min).

**Figure 18 polymers-17-00344-f018:**
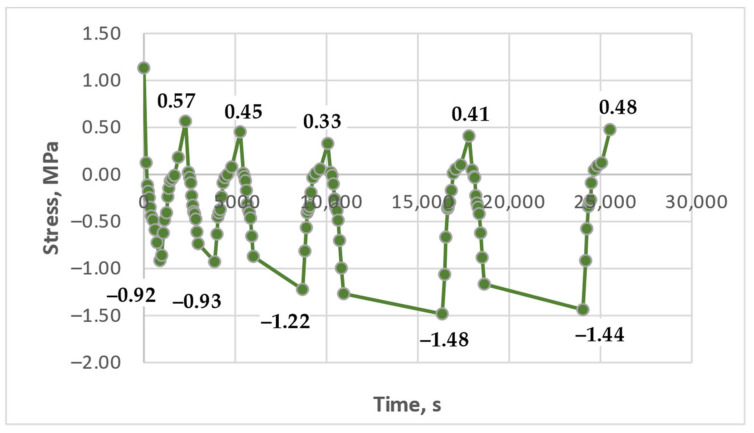
Dependence of stresses in the glass-reinforced plastic sample on time under cyclic thermomechanical loading (initial stress 1.13 MPa, heating to 130 °C, holds at 130 °C—2 min, 15 min, 45 min, 90 min, 90 min).

**Figure 19 polymers-17-00344-f019:**
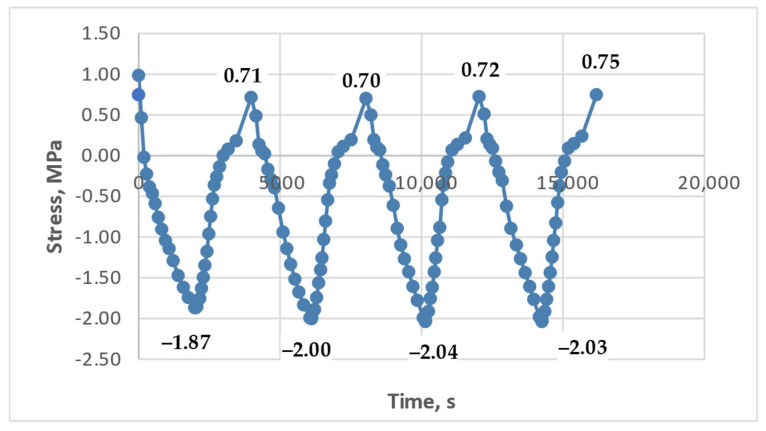
Dependence of stresses in the glass-reinforced plastic sample on time under cyclic thermomechanical loading (initial stress 1.0 MPa, heating to 180 °C, without holds).

**Figure 20 polymers-17-00344-f020:**
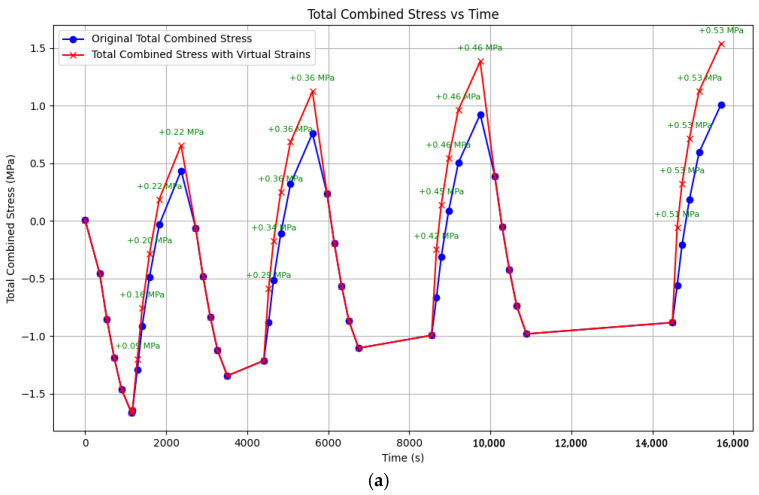
Modeling and experimental results on cyclic thermomechanical loading on epoxy polymer (heating to 80 °C, with holds, without applying initial tensile stresses): (**a**) model curves of stress changes over time during heating and cooling (blue—without considering additional residual stresses, red—with consideration); (**b**) experimental model curve of stress changes over time during heating and cooling.

**Figure 21 polymers-17-00344-f021:**
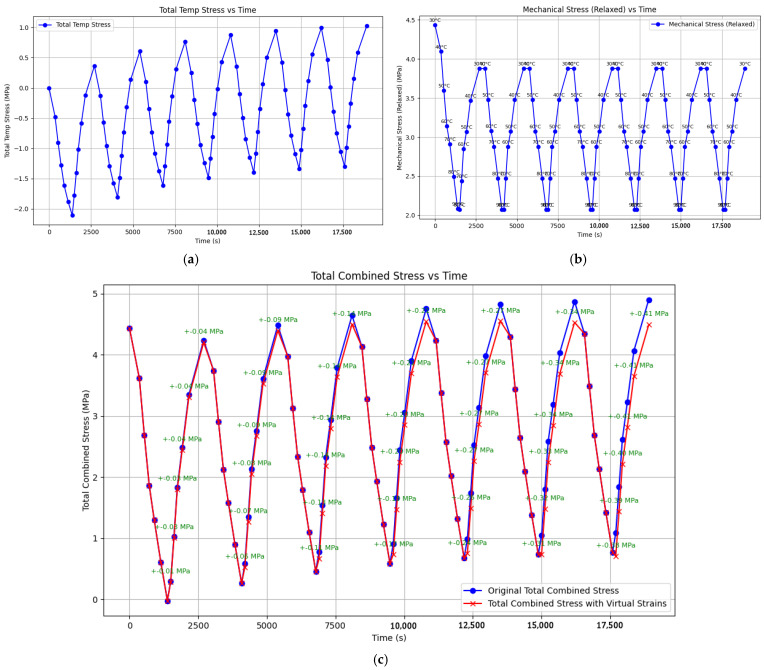
Results of modeling of cyclic thermomechanical impact on epoxy polymer (heating to 90 °C, without holding, initial tensile stresses 4.3 MPa): (**a**) curve of temperature stresses; (**b**) curve of mechanical stresses; (**c**) curves of total stresses (blue—without taking into account additional residual stresses, red—with taking into account).

**Figure 22 polymers-17-00344-f022:**
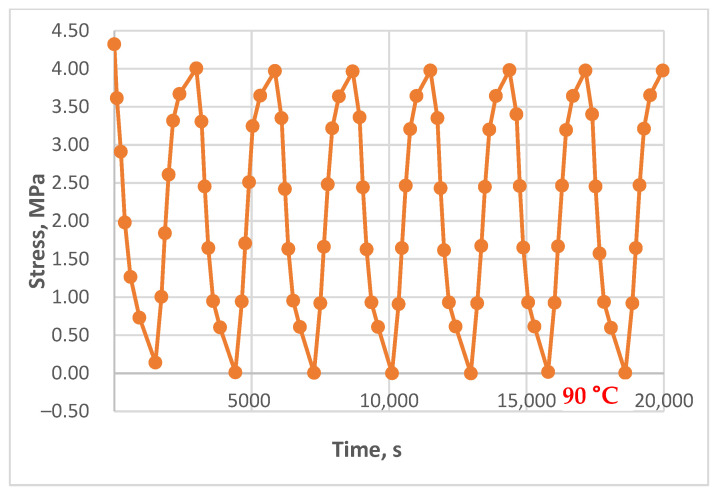
Experimental results for cyclic thermomechanical effect on epoxy polymer (heating to 90 °C, no holding time, initial tensile stresses of 4.3 MPa).

**Figure 23 polymers-17-00344-f023:**
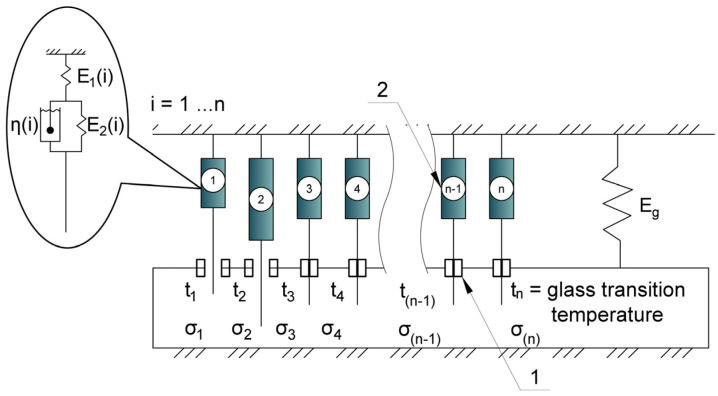
Schematic of a possible improved version of the model: 1—temperature brake, 2—cell equivalent to the Kelvin-Voigt three-element model.

**Table 1 polymers-17-00344-t001:** Results of parameter approximation E_1_, E_2_, and *η* based on the relaxation curve at different temperatures for the epoxy polymer at initial tensile stresses of 4.2 MPa.

	Model Relaxation Curve	Time Interval, s	E_1_, MPa	E_2_, MPa	*η*, MPa·s
**t = 30 °C** **σ_0_ = 4.2 MPa**	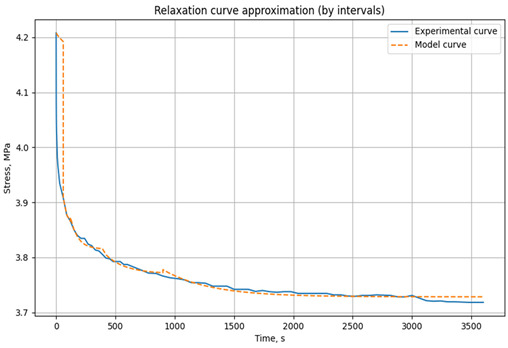	0–60	3045	381,810	3.57 × 10^7^
		60–120	3056	238,874	7.40 × 10^6^
		120–400	3046	228,618	1.45 × 10^7^
		** 400–900 **	** 3060 **	** 404,890 **	** 6.75 × 10^7^ **
		** 900–3600 **	** 3054 **	** 301,349 **	** 1.20 × 10^8^ **

**t = 60 °C** **σ_0_ = 4.2 MPa**	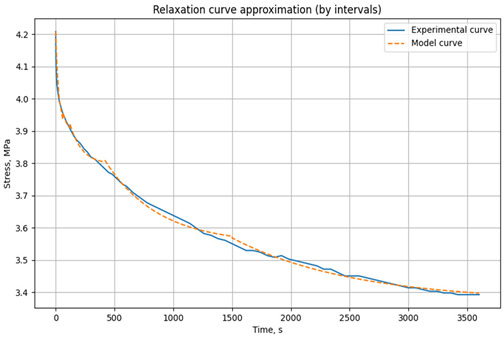	0–60	2783	35,982	1.00 × 10^6^
		60–120	2761	194,441	7.04 × 10^6^
		120–400	2764	94,170	9.87 × 10^6^
		400–1500	2779	42,637	2.04 × 10^7^
		1500–3600	2760	51,894	5.67 × 10^7^

**t = 90 °C** **σ_0_ = 4.2 MPa**	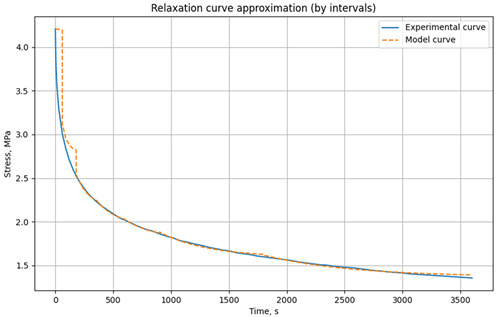	0–60	1658	315,760	2.55 × 10^7^
		60–90	1668	23,270	1.00 × 10^6^
		90–180	1661	10,338	1.26 × 10^6^
		180–360	1654	12,222	2.39 × 10^6^
		360–600	1658	13711	3.89 × 10^6^
		** 600–1500 **	** 1667 **	** 10,000 **	** 4.76 × 10^6^ **
		** 1500–3600 **	** 1668 **	** 10,000 **	** 8.63 × 10^6^ **


**Table 2 polymers-17-00344-t002:** Results of parameter approximation E_1_, E_2_, and *η* based on the relaxation curve at different temperatures for the glass-reinforced plastic at initial tensile stresses of 18.4 MΠa.

	Model/Experimental Relaxation Curve	Time Interval, s	E_1_, MPa	E_2_, MPa	*η*, MPa·s
**t = 30 °C** **σ_0_ = 5.6 MPa**	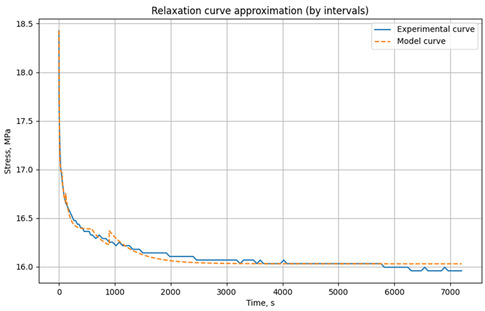	0–60	10,783	124,579	1.00 × 10^6^
		60–120	10,689	705,152	2.20 × 10^7^
		120–600	10,730	653,221	4.63 × 10^7^
		600–900	10,683	999,732	2.60 × 10^8^
		900–3600	10,773	774,976	3.64 × 10^8^

**t = 60 °C** **σ_0_ = 5.6 MPa**	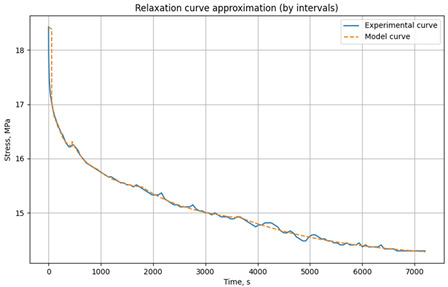				
		0–60	7827	348,725	1.82 × 10^8^
		60–90	7833	317,182	1.41 × 10^7^
		90–180	7821	314,093	3.64 × 10^7^
		180–360	7825	208,479	6.71 × 10^7^
		360–600	7810	303,202	1.42 × 10^8^
		600–1500	7814	174,597	1.97 × 10^8^
		1500–3600	7832	139,584	3.16 × 10^8^

**t = 90 °C** **σ_0_ = 5.6 MPa**	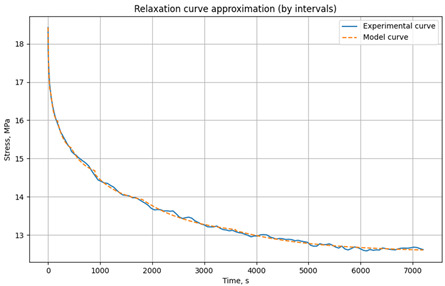	0–60	7105	65,605	1.00 × 10^6^
		60–90	7048	167,061	8.48 × 10^6^
		90–180	7101	143,522	1.97 × 10^7^
		180–360	7072	161,122	4.50 × 10^7^
		360–600	7091	128,993	6.95 × 10^7^
		600–1500	7077	100,500	1.12 × 10^8^
		1500–3600	7082	167,416	2.67 × 10^8^


## Data Availability

All data obtained in this paper are available from the authors upon request.

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
