# Peer review of "Viscoelastic Memory Effects in Cyclic Thermomechanical Loading of Epoxy Polymer and Glass-Reinforced Composite: An Experimental Study and Modeling Under Variable Initial Stress and Cycle Durations"

_polymers, 2025, doi:10.3390/polym17030344_

Round 1

Reviewer 1 Report

Comments and Suggestions for Authors

The authors studied the viscoelastic behavior of an epoxy polymer and a glass-reinforced composite under cyclic thermomechanical loading. They used a refined material model in Python to examine memory effects related to initial mechanical stresses and thermal cycles. The research identified viscoelastic parameters through the three-element Kelvin–Voigt model at various temperatures, providing insights into residual stress accumulation. Experimentation and modeling supported the findings, enhancing understanding of the stress-strain behavior of polymer-based composites in cyclic conditions. I highly recommend the publication of this manuscript after addressing the following minor concerns.

1. In Section 3.3, it is stated that the accumulation of residual stresses in the glass-reinforced composite is weaker than in the pure epoxy polymer. However, this is not clearly reflected in the modeling results in Section 3.4, where the composite's behavior is not discussed. The authors should explain this disparity or acknowledge the study's limitations in comparing the two materials.

2. The manuscript does not provide an in-depth explanation of the theoretical advantages of fractional-order derivative models in capturing memory effects. The authors should represent the theoretical background of memory effects and the advantages and limitations of fractional-order models briefly.

3. The figure title for Figure 18(b) is missing. The authors should make sure to include the accurate title for it.

4. The experiments were conducted at different temperature ranges for the epoxy polymer (30–90°C) and the glass-reinforced composite (30–180°C). The authors should provide a more explicit explanation for selecting these ranges, particularly why the polymer's temperature range is narrower, to ensure they accurately reflect real-world conditions.

5. In the relevant description regarding the generation and accumulation of residual stresses in epoxy polymer and glass reinforced composite under cyclic thermomechanical loading, please cite the following articles: Interfacial Si-O coordination for inhibiting the graphite phase enables superior SiC/Nb heterostructure joining by AuNi. https://doi.org/10.1016/j.compositesb.2024.111557. Tensile strain-mediated spinel ferrites enable superior oxygen evolution activity. https://doi.org/10.1021/jacs.3c08598.

6. Some grammar mistakes should be checked. Besides, the authors should better pay attention to the readability of text.

Reviewer 2 Report

Comments and Suggestions for Authors

In those sets of experimentations, the study is directed towards understanding and reporting the viscoelastic behavior of epoxy polymer and glass-reinforced composite under the condition of cyclic thermomechanical loading, particularly in regard to the formation and accumulation of residual stresses at given stress levels and thermal cycling conditions. The viscoelastic parameters are thus determined for the Kelvin-Voigt model using a Python-based material model having memory effects which describe the properties of materials at different temperatures, since this could provide certainty in the results. Presently, this shows that in comparison to glass-reinforced composite, resin has a larger stress accumulation residual depicted in its values that are attributed to the effect of reinforcement and stiffness. Furthermore, the experimental data from modelling results align well to support predictions with regard to polymer composite structures under cyclic loads.

Now, even though the introduction does talk about the importance of residual stresses and viscoelastic behavior, it doesn"t do a very good job of indicating where the investigation works to fill holes that exist in the current literature. For example, what limitations of existing models or experimental approaches does it address? 

It is put into the introduction that most applications disclose the use of epoxy polymers and glass-reinforced plastics in industries. What was the reason for including the materials in this study? Can their inherent attributes classify under a wider range of polymers and composites, or are they responding to specific industrial problems?

Recents works on GFRP composites for benefits and etc can be mentioned using: Effects of high pulling speeds on mechanical properties and morphology of pultruded GFRP composite flat laminates; The Effects of Eccentric Web Openings on the Compressive Performance of Pultruded GFRP Boxes Wrapped with GFRP and CFRP Sheets; Compressive Behavior of Pultruded GFRP Boxes with Concentric Openings Strengthened by Different Composite Wrappings

The introduction has quite an emphasis on the technical side and is not doing justice to the social value of the research outcomes. How would the results be likely to influence the design, safety, or life span of structures in civil engineering or as a whole? 

The introduction states what the work is about and what the tools are that have been deployed (e.g., Python scripts, Kelvin-Voigt model), but a more convincing justification would serve as a rationale: why was the three-element

The language of paper is not good. It should be corrected by a native. There are also russian words. Методика теоретических исследований

The results should be presented more better way. Currently it seems a part of msc thesis.

Comments on the Quality of English Language

The language of paper is not good. It should be corrected by a native. There are also russian words. Методика теоретических исследований

Reviewer 3 Report

Comments and Suggestions for Authors

1.        In the abstract part, it is suggested to remove relevant references. In addition, please provide some important quantitative results and conclusions found in this paper.

2.        The writing of the introduction section is too fragmented and contains a large number of paragraphs. It is recommended to merge multiple paragraphs into several paragraphs according to the research content of this paper. In addition, the authors have also emphasized the cycle temperature, aggressive environments and loading may have some significant effects on polymer resin composites, including residual stress, damage and failure, etc. However, these views should be supported by some recent research work, such as Construction and Building Materials, 2024, 440:137470. AIP Advances, 2023, 13(4).

3.        Please clarify the difference of viscoelastic thermomechanical behavior for polymers and composites. The difference between the two may be the fiber-resin interface, so please further clarify the effect of the interface on thermomechanical behavior.

4.        The introduction writing length is too long. In addition, you further summarize the problems solved and unsolved in the existing research work, and further highlighted some contributions of this paper.

5.        For GFRP samples, please provide the volume or mass fraction of the fiber and resin.

6.        The clarity of the pictures 3, 5 is very poor, it is recommended to replace them with some high definition pictures. At the same time, it is recommended to also check the unclear pictures in the full text.

7.        The topics of 2.2.3 are not in English. In addition, there are some non-English expressions in the main part of the paper. It is recommended to check the full text and carefully revise it, such as line 393-393, 743-747.

8.        For Figure 7 and Figure 8, for the exposure environment of 90 °C, it can be found that the change of stress over time continues to decrease. Therefore, can the time tested in this paper be sufficient to obtain the overall change of stress? Please add relevant explanation.

9.        Tables 1 ~ 2 have some related duplicates with Figures 7 and 8, further check is recommended.

10.    Some test processes in Part 3.2 are recommended to be placed in the Materials and Methods section. This part recommends only giving important results.

11.    The consistency between the simulation results and the experimental results should be further hinted.

12.    The conclusion part should be further enriched and including by 3 ~ 5 key points.

13.    The whole writing length of the paper is too long. In addition, some materials and test methods involved in the results part are suggested to be placed in Part 2.

Round 2

Reviewer 2 Report

Comments and Suggestions for Authors

The paper can be accepted

Reviewer 3 Report

Comments and Suggestions for Authors

Accept